# Towards Effective BIM/GIS Data Integration for Smart City by Integrating Computer Graphics Technique

**Junxiang Zhu** and **Peng Wu** *

School of Design and the Built Environment, Curtin University, Bentley 6102, Western Australia, Australia; junxiang.zhu@curtin.edu.au
* Correspondence: peng.wu@curtin.edu.au; Tel.: +61-8-9266-4723

**Abstract:** The development of a smart city and digital twin requires the integration of Building Information Modeling (BIM) and Geographic Information Systems (GIS), where BIM models are to be integrated into GIS for visualization and/or analysis. However, the intrinsic differences between BIM and GIS have led to enormous problems in BIM-to-GIS data conversion, and the use of City Geography Markup Language (CityGML) has further escalated this issue. This study aims to facilitate the use of BIM models in GIS by proposing using the shapefile format, and a creative approach for converting Industry Foundation Classes (IFC) to shapefile was developed by integrating a computer graphics technique. Thirteen building models were used to validate the proposed method. The result shows that: (1) the IFC-to-shapefile conversion is easier and more flexible to realize than the IFC-to-CityGML conversion, and (2) the computer graphics technique can improve the efficiency and reliability of BIM-to-GIS data conversion. This study can facilitate the use of BIM information in GIS and benefit studies working on digital twins and smart cities where building models are to be processed and integrated in GIS, or any other studies that need to manipulate IFC geometry in depth.

**Keywords:** Building Information Modeling (BIM); Geographic Information System (GIS); Industry Foundation Classes (IFC); 3D model; smart city; digital twin

## 1. Introduction

Smart city and digital twin require three-dimensional (3D) building models to constitute a large-scale city model, based on which analysis and decision-making processes regarding city management can be carried out [1]. This virtual city model serves as the frame of the digital representation of a physical city, to which other enabling technologies of the smart city, such as radio frequency identification (RFID) and real-time locating systems [2], can be attached. During the construction of such city models, a large number of building models are to be produced and linked via spatial locations [3]. Therefore, one technical requirement of the smart city and digital twin is the creation, management, and analysis of 3D building models. Many studies have suggested that Building Information Modeling (BIM) can be a fundamental technique in smart cities due to its strength in producing highly detailed building models, and Geographic Information Systems (GIS) can be prominent in managing and analyzing these models to a large extent via a global spatial reference system [4–6]. Therefore, the integration of BIM and GIS can be a fundamental technique for a smart city and digital twin, where building models produced by BIM are to be integrated into a GIS environment for visualization and analysis.

The integration of BIM and GIS is mainly conducted at two levels, i.e., application level and data level [7]. Data-level integration focuses on the data exchange between BIM and GIS, with the goal of achieving efficient information exchange between these two systems, which can facilitate application-level integration of BIM and GIS. Application-level integration, as the title suggests, explores the potential joint application of BIM and GIS to solve practical problems. For example, BIM and GIS have been jointly used in a variety of

applications related to the smart city, such as building-level flood damage assessment [8], low-energy building design [9], city fire emergence management [10], construction site layout optimization [11], and supply chain management [12].

BIM/GIS integration, as a technique enabling the smart city and digital twin, requires BIM data to be converted into a form/format that is accessible by GIS. There are mainly two common conversion paths available, involving Industry Foundation Classes (IFC), City Geography Markup Language (CityGML), and shapefile. IFC is widely used in the AEC domain for building information exchange and is the representative data standard for the BIM side. On the GIS side, CityGML and shapefile are commonly involved. CityGML is an international standard [13] and has attracted the attention of most researchers working on BIM/GIS integration, whereas shapefile is widely used in the geospatial industry. Shapefile is a native format of ArcGIS, the most frequently used GIS platform in BIM/GIS integration [14], but it is open to the community and supported by many open-source tools, such as QGIS. Accordingly, the two common conversion paths are the IFC-to-CityGML path and the IFC-to-shapefile path [7], both of which are important for BIM/GIS integration.

By far, data exchange, or data interoperability, between BIM and GIS is still a challenge [15], especially for the IFC-to-CityGML conversion. From the literature, it can be concluded that this challenge is intrinsically caused by the vast discrepancies between BIM and GIS in data creation, storage, and management [7,16]. These discrepancies have resulted in various data conversion tasks, some of which can be quite challenging, such as the solid-to-surface transformation required by the IFC-to-CityGML conversion [17].

With the emerging studies on the smart city and digital twin, there is a growing need for 3D building models, as well as a reliable and efficient approach for reconstructing or converting these models for their use in GIS. According to Biljecki et al. [3], 3D models have been applied in a variety of applications, from simple visualization to complex spatial analysis, such as indoor localization [18], indoor navigation [19–21], and room-level traffic noise assessment [22]. BIM can be a promising source of 3D building models for GIS. However, the data conversion problem mentioned above has become the bottleneck of using BIM models in GIS. Studies on the smart city and digital twin can be facilitated if the data conversion problem can be well addressed. Computer graphics techniques are promising in addressing this problem. Computer graphics deals with the display of graphics on computer screen, only using explicit points, edges, and faces [23]. Given the fact that IFC models containing implicit geometries can be eventually displayed, it is reasonable to assume that these implicit geometries have been converted in some way into explicit geometries by computer graphics techniques. If these points, edges, and faces generated by computer graphics techniques can be retrieved and interpreted, it is possible to preserve them in a format and use them in GIS.

The goal of this study is not to address the intrinsic differences between BIM and GIS in data format/standard but to facilitate the use of BIM models in a GIS environment and thus to benefit the development of the smart city and digital twin. This goal was achieved through two steps. In step one, a literature review was conducted to summarize the data conversion tasks involved in BIM-to-GIS data conversion, before the two common data conversion paths, i.e., IFC-to-CityGML and IFC-to-shapefile, were compared in terms of the number and difficulty of these conversion tasks. In step two, based on the previous comparison, a more efficient approach for data conversion was developed for the IFC-to-shapefile conversion by integrating a computer graphics technique.

## 2. Related Work

### 2.1. BIM-to-GIS Data Conversion

BIM models consist of geometric information and semantic information. Geometry provides information on the shape, size, and location of objects, while semantics provides information on the properties of objects, such as class type, material, and functions. These two types of information are essential to the integrity of BIM models. Accordingly, data

conversion from BIM to GIS usually involves two aspects, i.e., geometry conversion and semantics transfer [7,10,24].

Table 1 lists all the general tasks involved in BIM-to-GIS data conversion as well as corresponding studies working on the resolution of these problems. In general, the following tasks can be involved in data conversion: representation conversion, coordinate transformation, geo-referencing, model simplification, and semantics transfer [7,25,26].

**Table 1.** Tasks in data conversion from BIM to GIS.

| General Tasks | | Specific Tasks | |
|---|---|---|---|
| | | IFC-to-CityGML | IFC-to-Shapefile |
| Geometry | Representation conversion | Converting solid models to surface models (difficult): - Converting B-Rep [25] - Converting swept solid [25,27] - Converting CSG/Clipping [19,25] | Converting solid models to solid models: - Converting B-Rep [28] - Converting swept solid [10,28,29] - Converting CSG/Clipping [10,24,28] |
| | Coordinate transformation | Needed | Needed |
| | Geo-referencing | Needed, if to be integrated with other spatial data | Needed, if to be integrated with other spatial data [16,30,31] |
| | Model simplification | Needed, as Level of Detail (LoD) is defined - LoD1 [27,32–34] - LoD2 [27,32–34] - LoD3 [25,27,32,34] - LoD4 [27,32,34] | Optional |
| Semantics | Semantics transfer | Class mapping needed, as CityGML is a semantic data schema [27,34–38] | Semantics extraction needed, as shapefile is not a semantic data schema |

Representation conversion refers to the conversion of implicit Constructive Solid Geometry (CSG) and swept solid representations into the explicit Boundary Representation (B-Rep). This is required due to the difference in modeling paradigm between BIM and GIS [29]. Coordinate transformation refers to the transformation of coordinates of building elements from its local coordinate system to the global coordinate system of the IFC project. This is required due to the use of relative placement in IFC. Geo-referencing is another type of coordinate transformation, which transforms coordinates from the global coordinate system of the IFC project into a coordinate reference system that is related to the physical earth [30]. Geo-referenced BIM models can be integrated with other spatial datasets in GIS. Model simplification refers to the simplification of building models, which is required due to the additional storage space and rendering power required by over-detailed building models. These tasks belong to the category of geometry conversion.

Semantic information is another type of important information in BIM that should be properly transferred to GIS [7]. CityGML and shapefile tackle this task in different ways. CityGML is a semantic data model, which means that it defines classes for building components and the relationship between them; these classes have to be mapped with those from IFC in order to properly transfer semantic information. In contrast, shapefile is a more primitive data standard/format, in which class mapping is not required; instead, IFC semantics can be directly inherited and stored as attribute tables in shapefile.

### 2.2. IFC-to-CityGML Conversion

The IFC-to-CityGML conversion has to deal with more conversion tasks, and some of them are quite challenging in both geometry conversion and semantics transfer.

The geometry conversion for the IFC-to-CityGML path is more difficult than the IFC-to-shapefile path, as it involves the change of the modeling paradigm (from solid

modeling to surface modeling) and the conversion of Level of Detail (LoD). Detailed differences between the surface model and the solid model have been described in [39]. Many studies have attempted to address the geometry conversion issue. For example, in the study by Deng et al. [27], methods were developed to generate surfaces from IFC parameters for models from LoD1 to LoD4. Kang et al. [32,33] used the screen-buffer scanning-based multiprocessing (SB-MP) technique to generate LoD1 to LoD4 CityGML models, and Donkers et al. [25] developed an automatic approach for generating LoD3 CityGML models from IFC models by using a series of geometric operations, such as dilation and erosion.

In terms of semantics transfer, class mapping is a unique task that is mandatory for the IFC-to-CityGML conversion. A large amount of work has been carried out to address this problem by developing new data schemas or modifying current data schemas. For example, Deng et al. [27] used semantics from IFC and CityGML to establish the Semantic City Model. Karan et al. [35] used a semantic web technique to combine IFC and CityGML semantics. El-Mekawy et al. [36] developed a unified building model for converting IFC into CityGML. In addition, Application Domain Extensions (ADEs) can be developed for CityGML to receive additional semantic information from IFC [27,37,40,41].

The IFC-to-CityGML path has potential to be the standardized way for accommodating BIM information but is more difficult to realize. Despite the efforts mentioned above, it is still problematic in both geometry conversion and semantics transfer [42]. An easy-to-do and efficient approach for geometry conversion is still absent [16], not to mention that ADEs developed by various projects were project-specific and may not be recognized by some visualization tools [37]. This is probably the reason that CityGML was rarely used in studies on application-level BIM/GIS integration.

*2.3. IFC-to-Shapefile Conversion*

In contrast, the IFC-to-shapefile path is more workable for BIM-to-GIS data conversion, for four reasons. (1) First, there are fewer and easier conversion tasks in IFC-to-shapefile conversion. For example, the challenging solid-to-surface conversion and class mapping, which are mandatory for the IFC-to-CityGML conversion, are not required by the IFC-to-shapefile conversion. In this sense, the data conversion from BIM to GIS can be completed in an easier manner. (2) Second, behind shapefile are mature GIS systems, such as the prevalent ArcGIS. These systems have strong data management and analysis capacity that get shapefile ready for practical use, while CityGML models have to first be converted before they can be used in ArcGIS. (3) Third, in terms of shapefile itself, shapefile supports both solid models and surface models [43], which makes shapefile capable of accommodating 3D IFC geometry, and the relational database technique behind shapefile enables it to store, extend, and query IFC semantic information. (4) Fourth, shapefile is an open format widely used for geospatial data exchange. It has been adopted by researchers, industry, and governments, such as the Landgate of Western Australia [44] and Data.gov.au, which provides open government data in Australia [45]. All of these advantages make the IFC-to-shapefile path more realistic for the use of building models in GIS. This conversion path is, therefore, suggested by this study.

In contrast with the IFC-to-CityGML path, there are less studies on the IFC-to-shapefile conversion. Some commercial tools are available for this conversion, such as Feature Manipulation Engine (FME) and Data Interoperability extension for ArcGIS (DIA) [46]. DIA is toolset built on FME, which extends the ability of ArcGIS to read/write more than 100 formats in GIS and computer-aided design (CAD). They are then referred to as DIA/FME in this paper. These tools are able to convert IFC into shapefile. For example, FME has been used in several projects for model conversion, such as in studies by Amirebrahimi et al. [8,47,48] and Boyes et al. [49]. However, their commercial nature has limited their availability and it has also been noticed that they are not sufficiently reliable. DIA/FME can crash due to the incapability in processing some specific IFC representation types [49,50]. Moreover, the semantic information that DIA/FME can transfer is limited. These problems

have not been solved by the latest version of DIA/FME, which will be demonstrated later in this paper.

Several studies developed their own methods for converting IFC into shapefile, such as studies by Isikdag et al. [10,51] and Zhu et al. [24,28,29]. In the studies by Isikdag et al. [10,51], a package for creating shapefile was developed through an application programming interface (API) for shapefile. In the studies by Zhu et al. [24,28,29], an open-source approach (OSA) was developed to convert IFC geometry into shapefile. These studies realized the conversion of IFC to shapefile but failed to provide a reliable and efficient conversion approach. For example, the study by Isikdag [51] has the following problems: incorrect spatial orientation of transformed building models, long processing time, incomplete conversion of representations, and ineffectiveness in converting clipping geometry. The OSA developed by Zhu et al. [24,28,29] has solved some of these problems, such as incomplete conversion of representations and ineffectiveness in converting clipping geometry, but the overall efficiency of data conversion has not been well addressed due to the lack of an efficient and common way to convert implicit IFC geometries.

### 2.4. Computer Graphics Technique in BIM-to-GIS Data Integration

Concepts of representation types such as B-Rep and CSG used by IFC are originally from areas such as CAD, computational geometry, and computer graphics. These areas deal with the creation and visualization of 3D models using computers. From the perspective of computer graphics, explicit geometries are required for visualization, which requires those implicit geometries in IFC, such as swept solid and CSG, to be converted into explicit points, edges, and faces. This process is referred to as tessellation [52], triangulation [53], or model evaluation [39]. Tools such as Open CASCADE technology (OCCT) have been developed for this purpose. OCCT, which is an open-source software development kit used in the field of CAD, computer-aided manufacturing (CAM), and computer-aided engineering (CAE) [54–56], has been applied for manipulating and visualizing 3D models in a range of applications, such as CAD-based robot path planning and simulation [57] and the preparation of as-damaged models for post-earthquake BIM reconstruction [58].

Via a literature review, it has been noticed that this technique has been integrated into BIM-to-GIS data conversion [16,25,59]. For instance, OCCT was involved in data conversion as a part of IfcOpenShell, which is a common tool for parsing IFC files [25,49] and has been used by many applications, such as BIMserver [60]. For example, in Arroyo Ohori et al. [16], a data processing pipeline was developed for converting IFC into CityGML, where OCCT/IfcOpenShell was used to convert IFC geometry into a transitional format (Wavefront OBJ), which was later converted into CityGML. Zhao et al. [59] and Chen et al. [61] combined IFC and 3D tiles to create 3D visualization for building models via a web-based GIS system. In this system, OCCT/IfcOpenShell was used to convert IFC into Wavefront OBJ files as a part of data processing. Donkers et al. [25] used OCCT/IfcOpenShell to automatically generate LoD3 CityGML models. The data conversion pipelines in these studies have been generally presented in Figure 1.

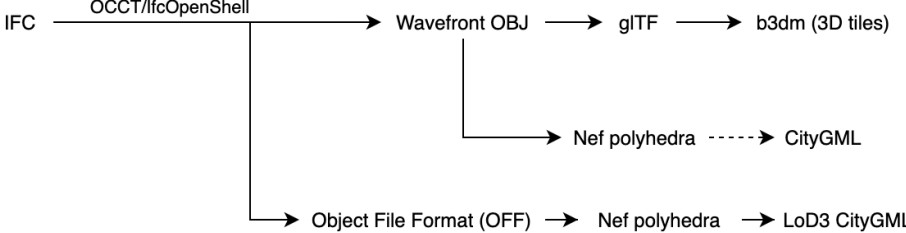

**Figure 1.** OCCT/IfcOpenShell in BIM-to-GIS data conversion by previous studies.

The above studies integrated OCCT into BIM-to-GIS data conversion, but the long data processing pipelines reflected their inefficiency in processing IFC data. These methods are not able to directly convert IFC into the destination format; instead, they had to first

convert IFC into a well-known intermediate format before they could be further processed. For example, Arroyo Ohori et al. [16] and Zhao et al. [59] had to first convert IFC files into OBJ files, then to Nef polyhedra, and finally to CityGML. This inefficiency in IFC data processing results in potential geometric information loss and a heavier data processing workload [61]. To be specific, the OBJ format used by Ohori et al. [16], Chen et al. [61], and Zhao et al. [59] as a transitional format can lead to serious geometric information loss, as geometries in an IFC file will be merged into a single shape after conversion [50]. In order to overcome this issue, Chen et al. [61], Zhao et al. [59], and Arroyo Ohori et al. [16] had to first split the IFC file into many sub-IFC files to ensure that each geometry was stored in a separate IFC file; these geometries were later combined again in the final format. This workaround not only introduced additional data processing work, but also caused additional data management problems, e.g., thousands of temporary sub-IFC files, or even more, can be generated in this process. These problems made this workaround vulnerable to errors and unsuitable to be used in smart city development, where a large number of complex building models can be involved.

This inefficiency is caused by the process-level use of OCCT, whose data exchange module can only export IFC into a finite number of formats, such as IGES, STL, OBJ, and VRML [62]. These formats are mainly used in the area of CAD and mainly focus on the geometric part of models, which made them not suitable for accommodating BIM information due to their insufficient support for semantics. The BIM-to-GIS data conversion can be more efficient and effective if computer graphics techniques, or OCCT, can be integrated into the conversion process at a lower level.

## 3. Materials and Methods

The rest of this paper mainly focuses on developing such a method that uses computer graphics techniques at a low level for converting IFC into shapefile. OCCT is adopted, as Arroyo Ohori et al. [16] found that OCCT is reliable in handling IFC models due to a lower geometry requirement, but other tools of this kind, such as the Computational Geometry Algorithms Library (CGAL), can also be used. The objective was then realized by two steps: (a) investigating OCCT (Section 3.1) and (b) integrating OCCT into the IFC-to-shapefile data conversion process (Section 3.2).

### 3.1. Investigating OCCT

3.1.1. Computer Graphics and OCCT

According to Eck [63], computer graphics is a broad field; it uses computational techniques to manipulate visual and geometric information and is closely related to fields such as computational geometry, computer vision, and applied mathematics. Computer-based systems, such as Autodesk Revit for BIM or ArcGIS for GIS, rely on computer graphics techniques to display geometric information on a computer screen. This visualization process is generally realized by four steps, as shown in Figure 2. (1) Application programs (e.g., Revit (California, United States) and ArcGIS (California, United States)) interpret and parse application data (e.g., IFC or shapefile); (2) the parsed data are converted into an intermediate format (or buffers) by application programs so that they can be processed by a Graphics Processing Unit (GPU) API (Application Programming Interface), such as OpenGL or WebGL [63]; (3) APIs pass buffers and corresponding commands to GPUs, and (4) GPUs process these data and transfer them to screens (display units) for the final visualization.

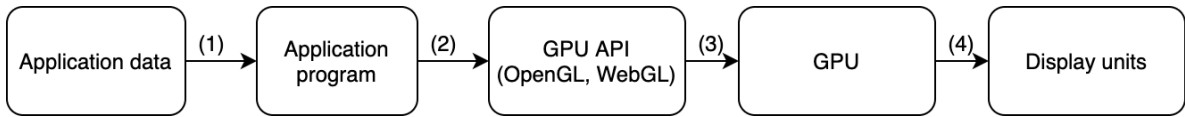

**Figure 2.** Information visualization on computer screen via computer graphics technique.

OCCT comprises several functional modules, each of which is designed for a specific purpose, such as model creation, geometry manipulation, visualization, and data exchange. For example, the data exchange module can read/write various CAD data formats and is what involved in the study by Arroyo Ohori et al. [16] for producing OBJ files. A detailed description of OCCT has been given in [62]. OCCT can be involved in the whole process of information visualization, but in this study, it mainly functions at step 1 and step 2 for reading and processing IFC geometries into buffers that are to be sent to the GPU API. Buffers can be sent to OpenGL in a desktop-based system or WebGL in a web-based system [63,64]. These buffers contain primitive geometric information that is to be displayed on a computer screen; therefore, extracting and interpreting these buffers is vital to this study.

### 3.1.2. Extracting and Converting IFC Geometry using OCCT

Understanding IFC, including its class hierarchy and spatial structure, is important for information extraction from IFC, especially for geometric information. For the purpose of data exchange in the AEC domain, IFC is developed and maintained by buildingSMART (formerly known as the International Alliance for Interoperability) as an open, software-neutral standard (ISO 16739-1:2018) [65–67]. In IFC, a variety of classes regarding building and related construction activities have been defined, among which only the *IfcProduct* class and its subclasses have geometric representation. This study mainly focuses on building element (*IfcBuildingElement*), which is a subclass of *IfcProduct*. Building elements are logically contained in a spatial structure element (*IfcSpatialElement*), such as a building story. Figure 3 presents the spatial structure elements and how they are linked with each other. The highest level of the spatial structure is assigned to *IfcProject*.

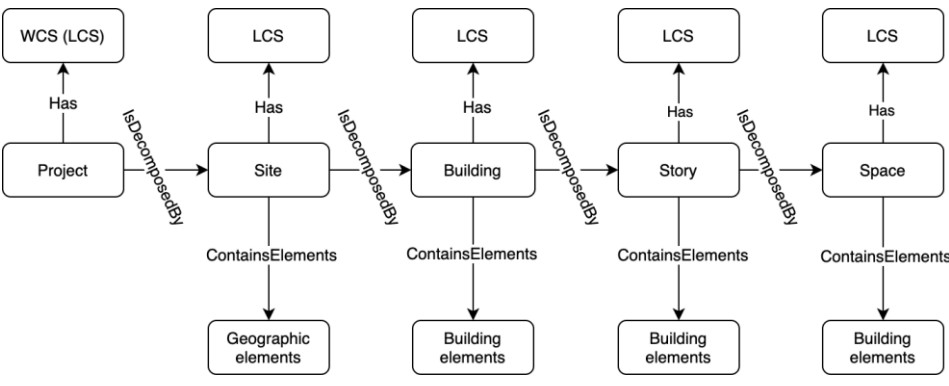

**Figure 3.** IFC spatial structure elements, i.e., site, building, building story, and space.

Building elements have various attributes, and the most important geometry-related attributes are representation (*IfcProductDefinitionShape*) and its placement (*IfcObjectPlacemen*). A representation determines the shape and size of an object, whereas the placement determines its location in the world coordinate system of the project. Figure 4 illustrates the data structure of a typical building element.

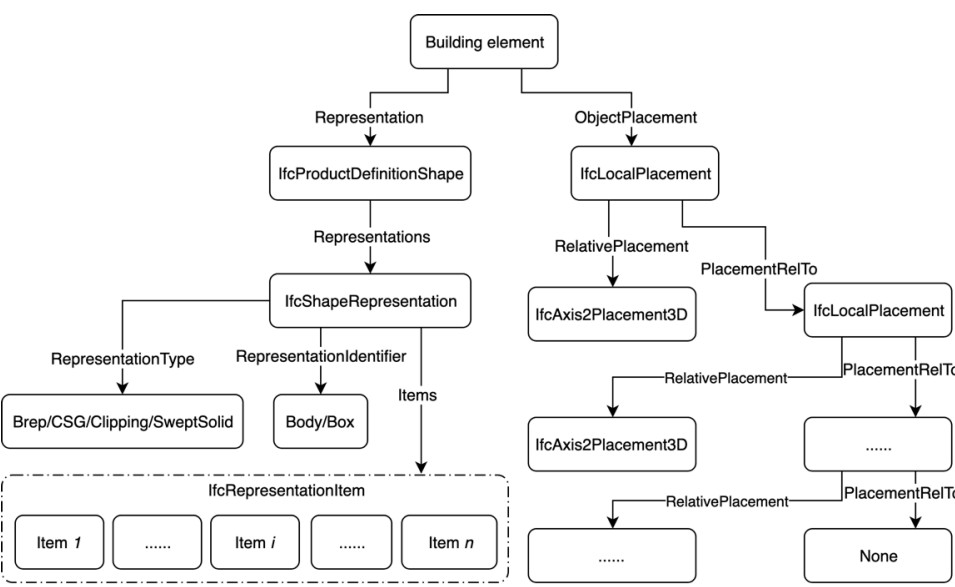

**Figure 4.** Representation and placement of a typical building element.

Based on the above two structures, building elements as well as their representations can be retrieved from IFC and converted using OCCT. The result of the conversion is a group of shape objects (buffers) that temporarily reside in the computer memory. These shape objects contain very primitive elements of geometry, referred to as triangulation elements in OCCT. They are originally supposed to be sent to the GPU for visualization [64]. These shape objects were closely examined in this study, and a method was developed to rebuild B-Rep from these primitive elements. The rebuilt B-Rep is referred to as OCCT B-Rep in this study.

### 3.1.3. From Primitive Triangulation Elements to OCCT B-Rep

The temporary shape objects contain a group of attributes, such as the unique identifier, name, and type that are inherited from IFC, and the most important one for rebuilding the geometric shape is the 'geometry', which has several attributes containing information on vertices, edges, and faces. These attributes are originally intended for machine processing, not human reading, but we managed to decipher them, as shown in Table 2.

**Table 2.** Interpretation of raw triangulation elements.

| | Raw Triangulation Elements | Interpreted Triangulation Elements |
|---|---|---|
| Shape.geometry.verts | $L1 : (x_1, y_1, z_1, \ldots, x_n, y_n, z_n)$ | $P : ((x_1, y_1, z_1), \ldots, (x_n, y_n, z_n))$ |
| Shape.geometry.edges | $L2 : (1, 2, 1, 3, \ldots)$ | $E : ((1, 2), (1, 3), \ldots)$ |
| Shape.geometry.faces | $L3 : (1, 2, 3, \ldots)$ | $F : ((1, 2, 3), \ldots)$ |

These attributes in Table 2 are essential for rebuilding geometric shape. The geometry.verts records a list of numbers ($L1$), which are the coordinates of all points in the shape, and every three numbers should be grouped to generate a list of points ($P$). The geometry.edges records another list of numbers ($L2$), which are the index of points, and every two numbers in the list should be grouped to generate a list of edges ($E$), where the first number indicates the start of an edge and the second indicates the end. The geometry.faces records a third list of numbers ($L3$), which are the index of points, and every three numbers should be grouped to generate a list of faces ($F$). From these deciphered variables, the geometric shape can be rebuilt using the method presented in Figure 5, where only verts and faces are used.

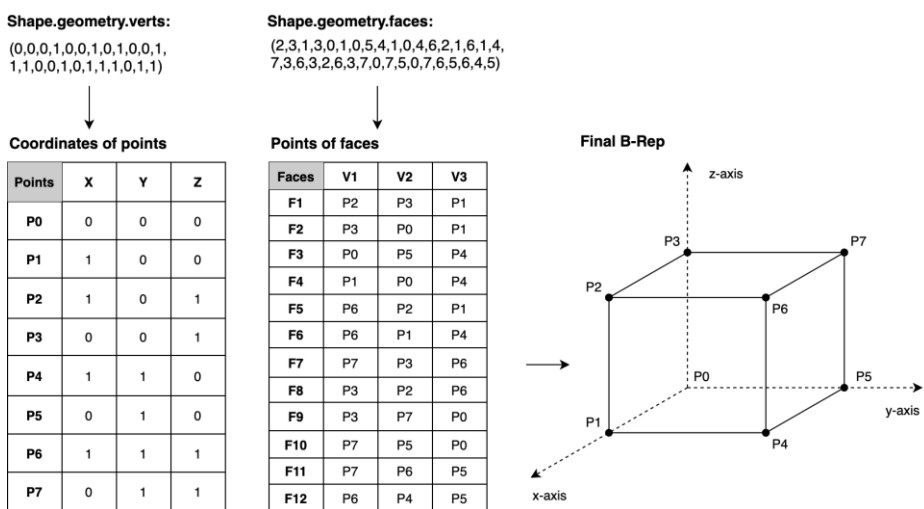

**Figure 5.** Converting triangulation elements into geometric shape.

### 3.2. Integrating OCCT into IFC-to-Shapefile Conversion

In order to integrate OCCT into IFC-to-shapefile data conversion, a method for converting OCCT B-Rep into shapefile B-Rep is required. In spite of the fact that both OCCT and shapefile use B-Rep to represent 3D objects, it is noticed during the investigation that they use different sub-types of B-Rep.

#### 3.2.1. Subtypes of B-Rep

In general, there are three sub-types of B-Rep depending on how points are organized into faces, including (a) explicit polygons (Type 1), (b) polygons defined by pointers into a point list (Type 2), and (c) explicit edges (Type 3) [23]. Table 3 shows the formats of these sub-types, in which $F$ stands for faces, $P$ stands for points, and $E$ stands for edges. (a) Explicit polygons (faces) are explicitly represented by a list of coordinates. (b) In Type 2, a face is defined by a list of pointers (or indexes) into the point list. For example, a face made up of points 1, 3, 5, and 7 in the point list is represented as $F : (1, 3, 5, 7)$. (c) In the third sub-type, a face is represented by a list of pointers into the edge list. For each edge, there are two pointers into the point list, as well as another one or two pointers for the face(s) to which the edge belongs.

**Table 3.** Sub-types of B-Rep, including explicit polygons (Type 1), polygons defined by pointers into a point list (Type 2), and explicit edges (Type 3).

| B-Rep Sub-Type | B-Rep Description |
|---|---|
| Type 1 | $F : ((x_1, y_1, z_1), (x_2, y_2, z_2), \ldots, (x_n, y_n, z_n))$ |
| Type 2 | $P : ((x_1, y_1, z_1), \ldots, (x_n, y_n, z_n))$ <br> $F : (1, 3, 5, 7)$ |
| Type 3 | $P : ((x_1, y_1, z_1), \ldots, (x_n, y_n, z_n))$ <br> $E : (V_1, V_2, P_1, P_2)$ <br> $F : (E_1, \ldots, E_n)$ |

#### 3.2.2. Converting OCCT B-Rep to Shapefile B-Rep

The actual B-Rep formats used by OCCT and shapefile are slightly different from the three general sub-types described above. The OCCT B-Rep has been introduced in the previous section, while the shapefile B-Rep is close to Type 1, and the difference is that, in shapefile B-Rep, for each face, the first point and the last point must coincide in order to form a closed ring [43], as follows:

$$F : ((x_1, y_1, z_1), \ldots, (x_n, y_n, z_n), (x_1, y_1, z_1)). \tag{1}$$

When generating shapefile B-Rep, there are two concerns: (1) points of faces and (2) orientation of faces, which is determined by the order of points in the face. By checking shape objects generated by OCCT, it is confirmed that the second concern has not been addressed. OCCT generates B-Rep where points are counter-clockwise relative to an outside observer (see Figure 6a), whereas the order of points in shapefile B-Rep should be clockwise [68,69] (see Figure 6b). Therefore, when converting OCCT B-Rep to shapefile B-Rep, the order of points has to be reversed.

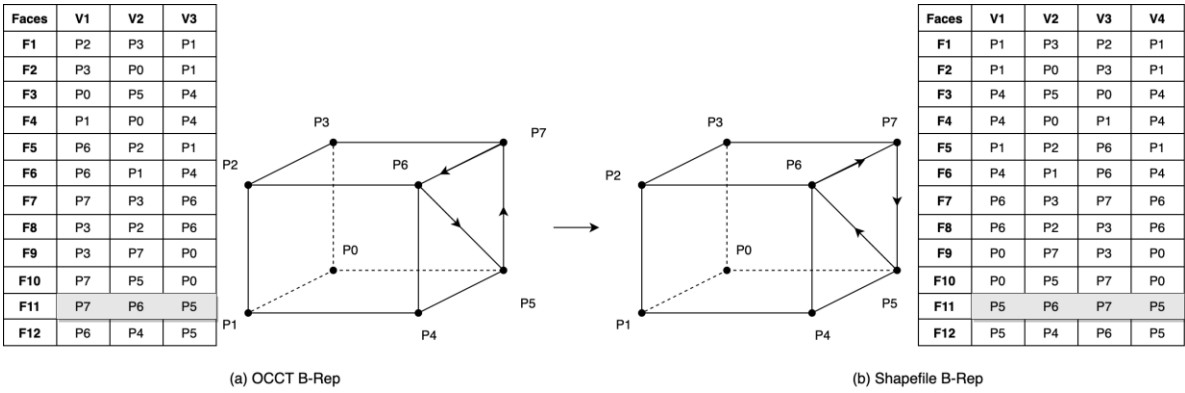

**Figure 6.** Reversing order of points for converting OCCT B-Rep to shapefile B-Rep.

The root of this problem is found to be in the fact that the DirectX, which is the default GPU API used in ArcGIS by ESRI (i.e., the developer of the shapefile format), uses a left-handed coordinate system, while the GPU API used by OCCT, i.e., OpenGL, uses a right-handed coordinate system. Despite the difference in point order, the normal of face (or the positive side of face) is pointing outward in the corresponding system (see Figure 7), which is important for calculating surface area, solid volume, and model rendering [39].

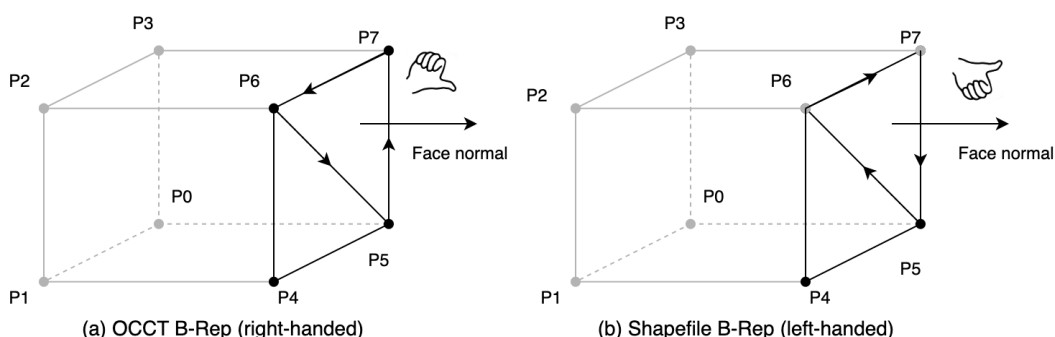

**Figure 7.** Normal of face (outward) in OCCT B-Rep and shapefile B-Rep.

The algorithm for converting OCCT B-Rep into shapefile B-Rep is graphically presented in Figure 8. In this algorithm, an object from IFC is first converted using OCCT into temporary shape objects. Then, from the shape objects, a point list ($P$) is derived from the coordinate list ($L1$), and a face list ($F$) is obtained from the point index list for face ($L3$), and, finally, the shapefile B-Rep is generated from the point list ($P$) and the face list ($F$).

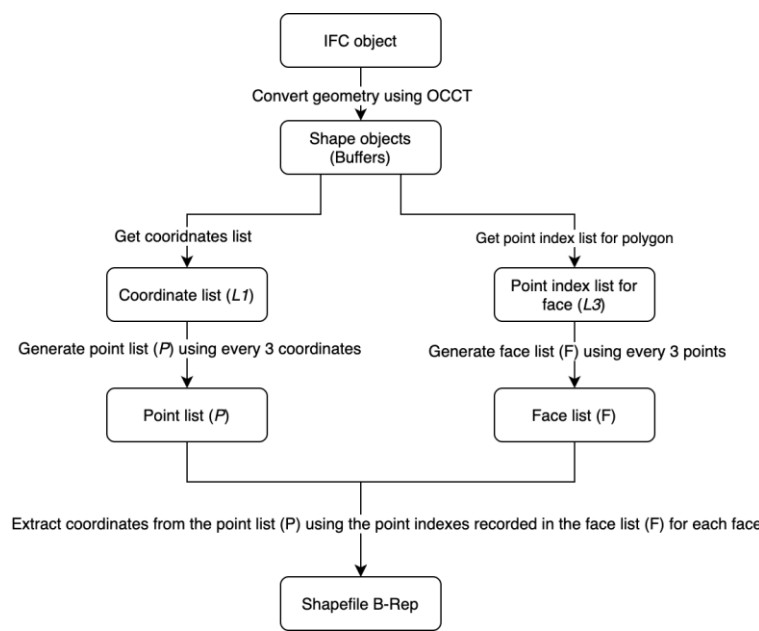

**Figure 8.** Converting OCCT B-Rep to shapefile B-Rep.

The key Python codes for geometry conversion and coordinate transformation are presented in Table 4. The *OCCT_shape* refers to the shape objects, the *objectPlacement* is the placement of object, and the *keepTransform* is a customized function for coordinate transformation using parameters directly from IFC. The equation behind the function is given by Zhu et al. [29] as follows:

$$[x'\ y'\ z'] = [x\ y\ z] \times \left( \left[ \vec{x}\ \vec{y}\ \vec{z} \right]^{\mathbf{T}^{-1}} \right)^{\mathbf{T}} + [x_1\ y_1\ z_1] \tag{2}$$

where $[x'\ y'\ z']$ is the transformed coordinates; $[x\ y\ z]$ is the initial coordinates; $[x_1\ y_1\ z_1]$ denotes the origin shift; $\vec{x}$, $\vec{y}$ and $\vec{z}$ are three perpendicular unit vectors indicating the direction of the *x*-axis, *y*-axis, and *z*-axis, respectively.

**Table 4.** Python codes for geometry conversion and coordinate transformation.

```
def OCCTToShapefile(OCCT_shape, objectPlacement):
    parts = [[]
    raw_verts = OCCT_shape.geometry.verts
    raw_faces = OCCT_shape.geometry.faces
    points = [raw_verts[i:i+3] for i in range(0, len(raw_verts),3)]
    faces = [raw_faces[i:i+3] for i in range(0,len(raw_faces),3)]
    points = keepTransform(points, objectPlacement)
    points = np.mat(points).tolist()
    for face in faces:
        ring = [points[face[2]], points[face[1]], points[face[0]], points[face[2]]]
        parts.append(ring)
    return parts
```

Using this algorithm, the example given in Figure 5 can be converted (see Figure 9). Rings are included in the 'parts' variable in Table 4. From this variable, together with the type of part (ring, indicated by code 5) and the type of shapefile (multipatch, indicated by code 31), shapefiles can be eventually generated. In shapefile, multipatch geometry is stored using several attributes, including multipatch.points, multipatch.z, and multipatch.parts. Among them, multipatch.points records the x and y coordinates, multipatch.z records the

z coordinates, while multipatch.parts records the index of points for the start and end of each ring.

**Shape.geometry.verts:**

(0,0,0,1,0,0,1,0,1,0,0,1,
1,1,0,0,1,0,1,1,1,0,1,1)

→

**Shape.geometry.faces:**

(2,3,1,3,0,1,0,5,4,1,0,4,6,2,1,6,1,4,
7,3,6,3,2,6,3,7,0,7,5,0,7,6,5,6,4,5)

| Faces | Rings (Parts) | Part type | Shape type |
|---|---|---|---|
| F1 | [ [1, 0, 0], [0, 0, 1], [1, 0, 1], [1, 0, 0] ] | Ring (5) | Multipatch (31) |
| F2 | [ [1, 0, 0], [0, 0, 0], [0, 0, 1], [1, 0, 0] ] | Ring (5) | Multipatch (31) |
| F3 | [ [1, 1, 0], [0, 1, 0], [0, 0, 0], [1, 1, 0] ] | Ring (5) | Multipatch (31) |
| F4 | [ [1, 1, 0], [0, 0, 0], [1, 0, 0], [1, 1, 0] ] | Ring (5) | Multipatch (31) |
| F5 | [ [0, 0, 0], [1, 0, 1], [1, 1, 1], [0, 0, 0] ] | Ring (5) | Multipatch (31) |
| F6 | [ [1, 1, 0], [1, 0, 0], [1, 1, 1], [1, 1, 0] ] | Ring (5) | Multipatch (31) |
| F7 | [ [1, 1, 1], [0, 0, 1], [0, 1, 1], [1, 1, 1] ] | Ring (5) | Multipatch (31) |
| F8 | [ [1, 1, 1], [1, 0, 1], [0, 0, 1], [1, 1, 1] ] | Ring (5) | Multipatch (31) |
| F9 | [ [0, 0, 0], [0, 1, 1], [0, 0, 1], [0, 0, 0] ] | Ring (5) | Multipatch (31) |
| F10 | [ [0, 0, 0], [0, 1, 0], [0, 1, 1], [0, 0, 0] ] | Ring (5) | Multipatch (31) |
| F11 | [ [0, 1, 0], [1, 1, 1], [0, 1, 1], [0, 1, 0] ] | Ring (5) | Multipatch (31) |
| F12 | [ [0, 1, 0], [1, 1, 0], [1, 1, 1], [0, 1, 0] ] | Ring (5) | Multipatch (31) |

**Multipatch.points:**  [(1,0),(0,0),(1,0),(1,0),(1,0),(0,0)(0,0),(1,0),(1,1),(0,1),(0,0),(1,1),(1,1),(0,0),(1,0),(1,1),(0,0),
(1,0),(1,1),(0,0),(1,1),(1,0),(1,1),(1,1),(1,1),(0,0),(0,1),(1,1)(1,1),(1,0),(0,0),(1,1),(0,0),(0,1),
(0,0),(0,0),(0,0),(0,1),(0,1),(0,0),(0,1),(1,1),(0,1),(0,1),(0,1),(1,1),(1,1),(0,1)]

**Multipatch.z:**  [0,1,1,0,0,0,1,0,0,0,0,0,0,0,0,1,1,0,0,0,1,0,1,1,1,1,1,1,1,1,0,1,1,0,0,0,1,0,0,1,1,0,0,0,1,0]

**Multipatch. parts:**  [0,4,8,12,16,20,24,28,32,36,40,44]

**Figure 9.** Converting OCCT B-Rep into shapefile B-Rep.

### 3.3. Semantic Information Transferring

Semantic information in this study refers to non-geometric information of BIM models, such as the type of an object, the relationship between objects, and other types of attributes defined by relationship entities. IFC has defined a large number of attributes for each entity through relationship entities, such as *IfcRelDefinesByProperties* and *IfcRelAssociatesMaterial*. It is possible to extract all attributes from an IFC file; however, in practice, only a small portion of these attributes are needed by a specific project [70].

In this study, the following essential attributes of building components, defined in the Building Topology Ontology [71], were extracted, including IFC class name, story level, building, and site. The Python codes for retrieving these attributes have been presented in Appendix A. These attributes are sufficient to answer questions such as what and where an object is. In addition, the unique identifier of each building component was retained, so that in cases where semantics needs to be extended, additional attributes can be extracted from IFC and linked with the model. This is fully supported by the relational database technique behind shapefile and facilitated by the fact that the vast property sets defined in IFC are in the form of tables. An example is provided in Figure 10. The benefit of transferring semantics in this way is keeping the converted building models in a compact manner but retaining the possibility of semantics extension. This feature is an advantage of shapefile over those CAD-oriented 3D formats, such as OBJ. Please note that rigid semantics mapping, which is mandatory for IFC-to-CityGML conversion, is not required by IFC-to-shapefile conversion, which further simplifies the process of semantics transfer.

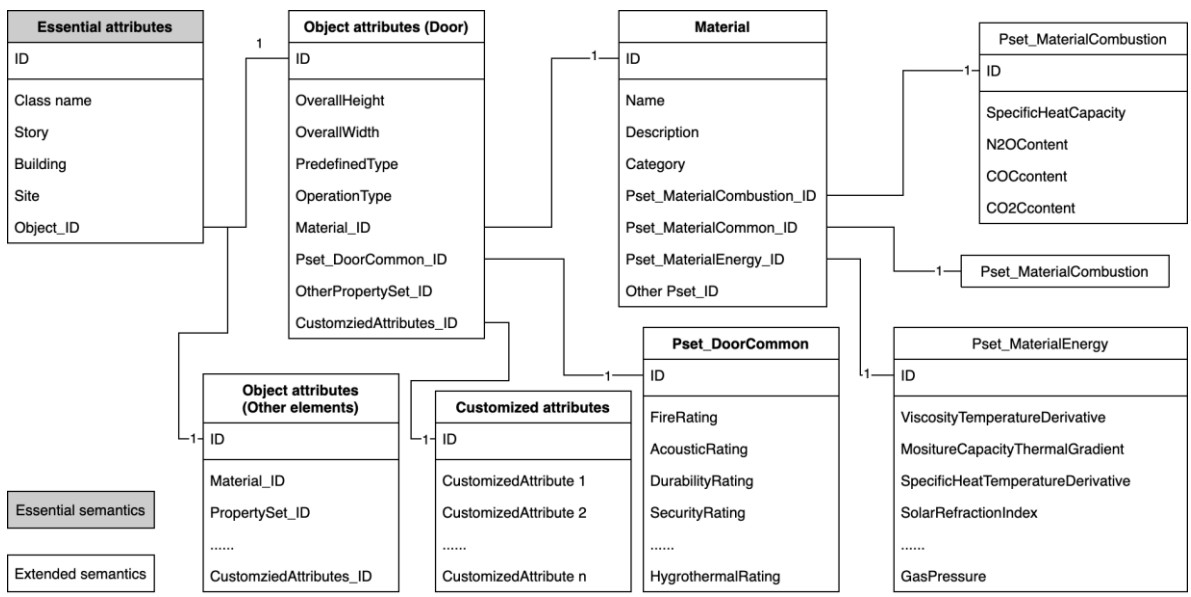

**Figure 10.** Essential semantics and extended semantics.

### 3.4. An Overview of the Proposed Approch for IFC-to-Shapefile Conversion

An overview of the final approach for data conversion from IFC to shapefile is given in Figure 11. The geometric information and semantic information are converted/transferred separately into a .shp file and .dbf file and linked via the .shx file, which are the three main files of shapefile.

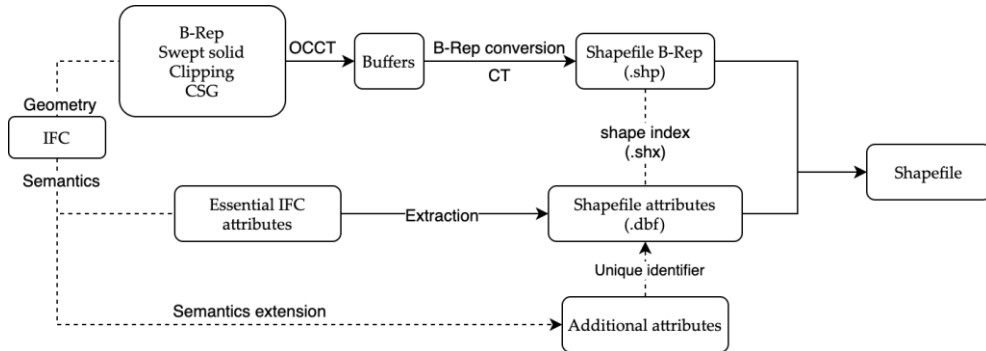

**Figure 11.** Overview of the proposed approach for IFC-to-shapefile conversion.

In this process, OCCT converts IFC objects into temporary shape objects (buffers); these shape objects are interpreted into OCCT B-Rep and converted into shapefile B-Rep using the proposed method. Some algorithms developed in OSA were also used in this study, such as those for coordinate transformation (CT) and shapefile creation. This approach is thus referred to as OCCT-OSA in this study, for the purpose of an easier discussion. The generated building models can be managed in two ways, i.e., either using a single shapefile for all the building elements or one shapefile for each building element. While both of these two ways can be realized, in this study, the first strategy was adopted, for easier data management.

### 3.5. Data

Four publicly available BIM models from the Karlsruhe Institute of Technology (KIT) and Open IFC Repository have been used to develop and validate the proposed method, including two house models and two building models. Figure 12 shows these BIM models,

including (a) House 1, (b) Institute, (c) House 2, and (d) Smiley. House 1, Institute, and Smiley are from KIT [72]; they were created as examples of quality IFC models. House 2 is a model from the Open IFC Model Repository (OIMR) [73]. Detailed information about these models can be found in [72]. Table 5 lists the type and quantity of building elements in each building model.

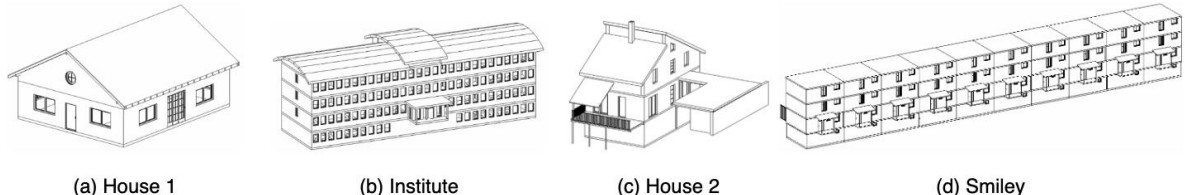

(a) House 1    (b) Institute    (c) House 2    (d) Smiley

**Figure 12.** BIM models used in this study for validation.

**Table 5.** Quantity of building elements in each model.

| IFC Class | Quantity of Components | | | |
|---|---|---|---|---|
| | **House 1** | **House 2** | **Institute** | **Smiley** |
| IfcBeam | 4 | 39 | - | 10 |
| IfcBuildingElementProxy | - | 8 | - | - |
| IfcColumn | - | 10 | 2 | 20 |
| IfcDoor | 5 | 14 | 77 | 170 |
| IfcMember | 42 | - | - | - |
| IfcRailing | 2 | 6 | 12 | 120 |
| IfcRoof | - | 1 | - | - |
| IfcSlab | 4 | 8 | 26 | 120 |
| IfcStair | 1 | 4 | 4 | 30 |
| IfcWall | - | 57 | - | 281 |
| IfcWallStandardCase | 13 | 46 | 121 | 270 |
| IfcWindow | 11 | 25 | 206 | 80 |
| | 82 | 172 | 448 | 831 |

## 4. Results

These four models were first assessed using OCCT alone to ensure that they could be processed and visualized. The following steps were used: (a) extracting shape representations of building elements from IFC, (b) processing these shape representations into shape objects, (c) sending shape objects to Open GL, and (d) creating a window on the screen to display the processed model. The result of this assessment is presented in Figure 13, where models are preliminarily rendered using random colors.

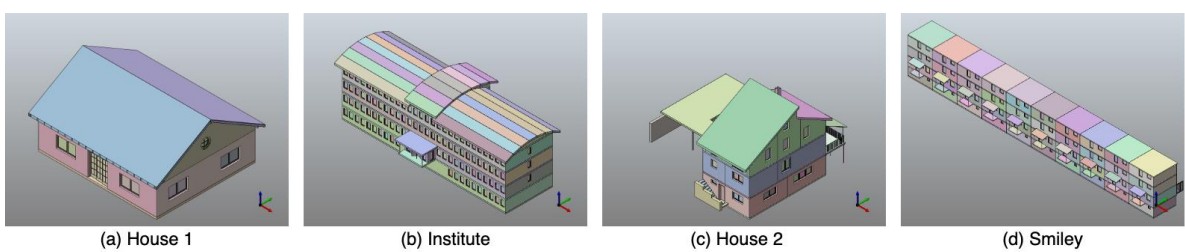

(a) House 1    (b) Institute    (c) House 2    (d) Smiley

**Figure 13.** Models processed and visualized using OCCT.

After the initial test, these four building models were then processed using the proposed approach and the result shows that they can be successfully converted into shapefile models. Figure 14 presents the converted shapefile models for (a) House 1, (b) Institute, (c) House 2, and (d) Smiley, including exterior, interior, indoor space (if defined), and an example floor plan for the ground floor. These models were visualized using ArcScene.

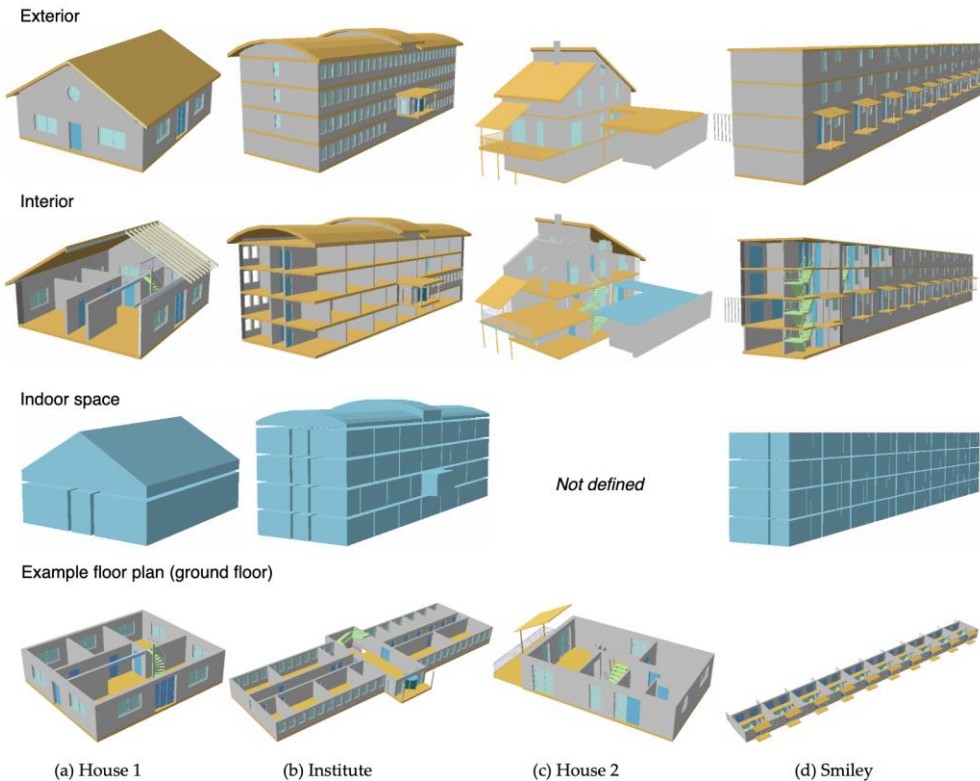

**Figure 14.** Converted building models, i.e., (**a**) House 1, (**b**) Institute, (**c**) House 2, and (**d**) Smiley.

Tables 6 and 7 present the processing time, file size, and quantity of building components before and after conversion. From the quantity of components, it can be asserted that the geometry of all building elements and building components have been retained. The converted building components for those models have been presented in Figure 15.

**Table 6.** Processing time and file size of converted models.

|  | House 1 | House 2 | Smiley | Institute |
|---|---|---|---|---|
| Time (s) | 8.5 | 4.6 | 55.2 | 17.9 |
| Size (KB) | 2878.2 | 1640.6 | 24097.2 | 5318.0 |

**Table 7.** Quantity of building components before and after conversion.

| IFC Class | House 1 | | House 2 | | Institute | | Smiley | |
|---|---|---|---|---|---|---|---|---|
|  | **Before** | **After** | **Before** | **After** | **Before** | **Converted** | **Original** | **Converted** |
| IfcBeam | 4 | 4 | 39 | 39 | - | - | 10 | 10 |
| IfcBuildingElementProxy | - | - | 8 | 8 | - | - | - | - |
| IfcColumn | - | - | 10 | 10 | 2 | 2 | 20 | 20 |
| IfcDoor | 5 | 5 | 14 | 14 | 77 | 77 | 170 | 170 |
| IfcMember | 42 | 42 | - | - | - | - | - | - |
| IfcRailing | 2 | 2 | 6 | 6 | 12 | 12 | 120 | 120 |
| IfcRoof | - | - | 1 | 1 | - | - | - | - |
| IfcSlab | 4 | 4 | 8 | 8 | 26 | 26 | 120 | 120 |
| IfcStair | 1 | 1 | 4 | 4 | 4 | 4 | 30 | 30 |
| IfcWall | - | - | 57 | 57 | - | - | 281 | 281 |
| IfcWallStandardCase | 13 | 13 | 46 | 46 | 121 | 121 | 270 | 270 |
| IfcWindow | 11 | 11 | 25 | 25 | 206 | 206 | 80 | 80 |
| Total quantity | 82 | 82 | 172 | 172 | 448 | 448 | 831 | 831 |

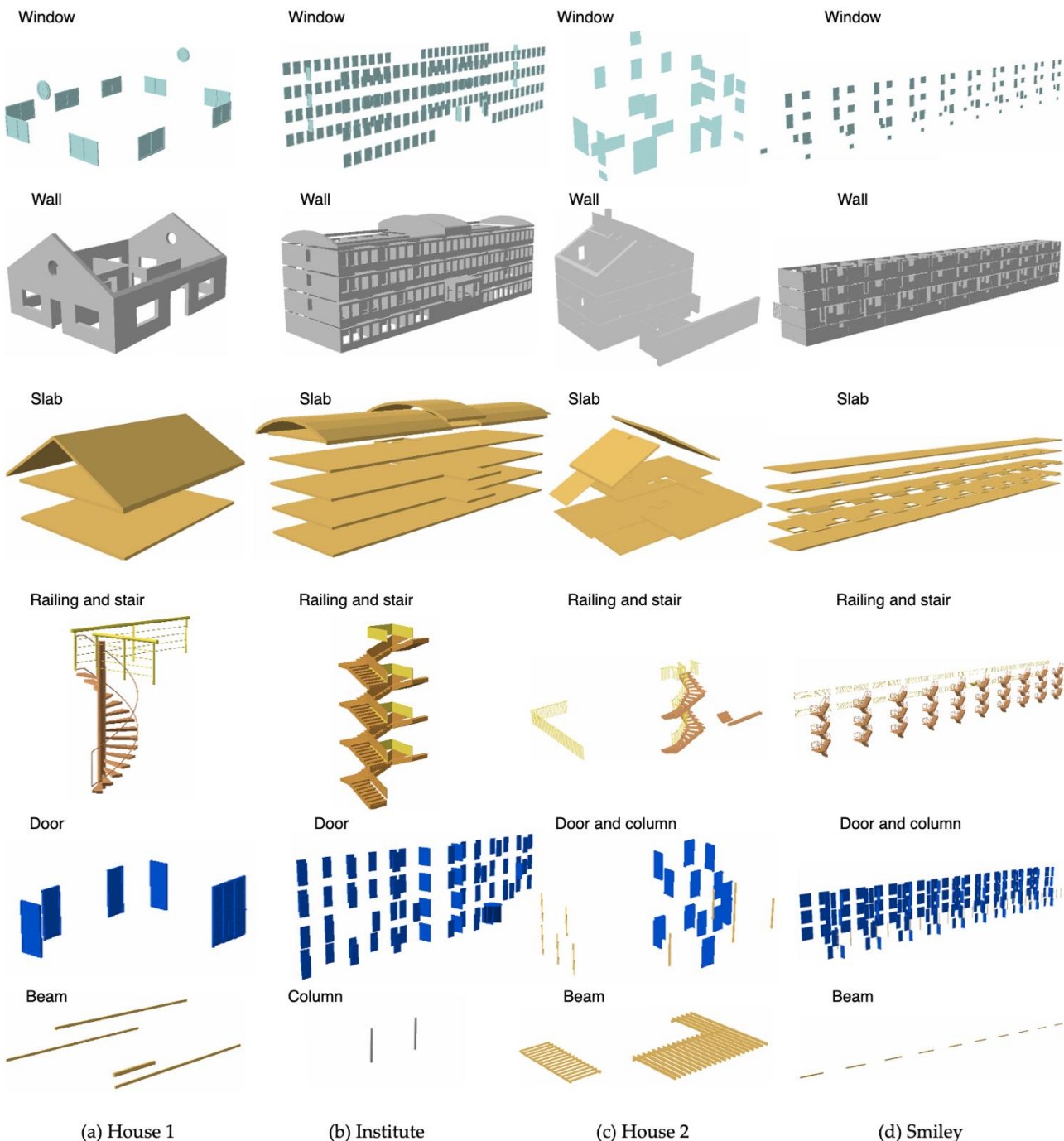

**Figure 15.** Converted building components in shapefile for (**a**) House 1, (**b**) Institute, (**c**) House 2, and (**d**) Smiley.

The original building components in IFC are presented in Figure 16, which were visualized using FZKViewer. The comparison between the converted building components and the original building components shows that all building components have been generated. Some doors and windows of these models in Figure 16 are open, which is a feature of the FZKViewer that can show the status of doors and windows. When visualized using Autodesk Revit, these doors and windows are closed.

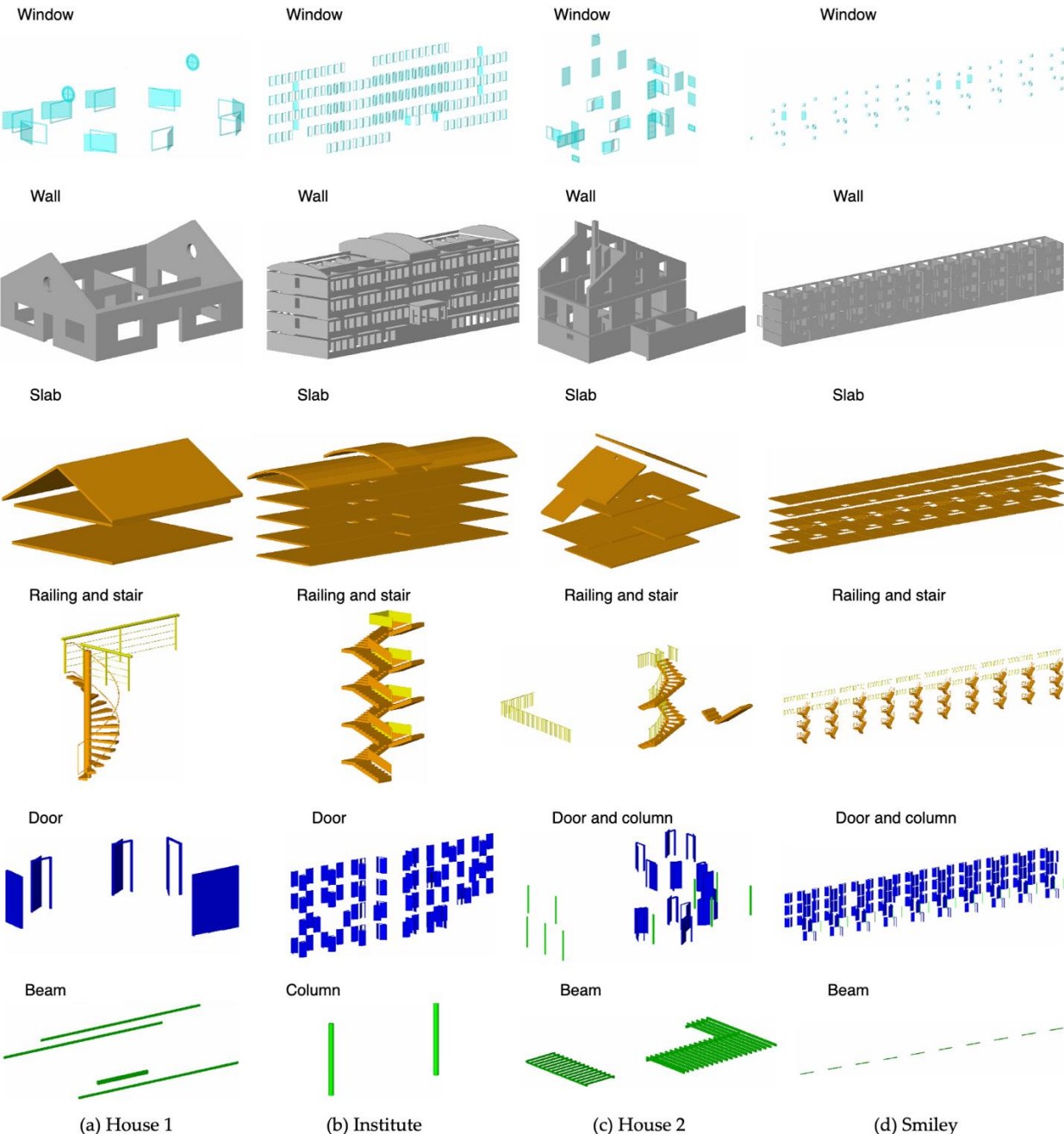

**Figure 16.** Building components in IFC for (**a**) House 1, (**b**) Institute, (**c**) House 2, and (**d**) Smiley.

## 5. Discussion

### 5.1. Comparing OCCT-OSA with Previous Methods

In order to assess the performance of the proposed method, it is compared with the OSA and the prevalent commercial tool, i.e., DIA/FME, in terms of processing time and file size of the converted models. The efficiency of data conversion is assessed using processing time. A longer processing time indicates a lower efficiency.

#### 5.1.1. Comparison between OCCT-OSA and OSA

(1) Efficiency of data conversion. Table 8 presents the comparison in processing time between OCCT-OSA and OSA as well as the improvement in efficiency.

**Table 8.** Processing time by OCCT-OSA and OSA.

| | Processing Time (Seconds) | | | | |
|---|---|---|---|---|---|
| | **House 1** | **House 2** | **Smiley** | **Institute** | **Average** |
| OSA (a) | 25.2 | 40.7 | 245.9 | 90.5 | - |
| OCCT-OSA (b) | 8.5 | 4.6 | 55.2 | 17.9 | - |
| Improvement (a-b)/b | 196% | 785% | 346% | 406% | 433% |

The result shows that OCCT-OSA took less time than OSA to process all these models, which indicates that OCCT-OSA has a higher efficiency. In the House 2 model, for example, OCCT-OSA only used 4.6 s, while OSA used 40.7 s, indicating an efficiency increase by 785%. In this study, the efficiency of data conversion has increased by between 196% and 785% and, on average, by 433%.

(2) File size. Table 9 presents the comparison between OCCT-OSA and OSA in file size. OCCT-OSA generated models with larger file sizes for these models. For example, the file size of House 1 from OCCT-OSA is 2878.2 KB, which is almost twice that of the model generated by OSA. This is due to the fact that OCCT-OSA generates more geometric details—for example, for doors and windows, as shown in Figure 17. On average, models generated by OCCT-OSA use 68% more storage space than OSA.

**Table 9.** File size of models generated by OCCT-OSA and OSA.

| | File Size (KB) | | | | |
|---|---|---|---|---|---|
| | **House 1** | **House 2** | **Smiley** | **Institute** | **Average** |
| OSA (a) | 1503.6 | 1120.0 | 14,305.5 | 3206.4 | - |
| OCCT-OSA (b) | 2878.2 | 1640.6 | 24,097.2 | 5318.0 | - |
| Comparison (b-a)/a | 91% | 46% | 68% | 66% | 68% |

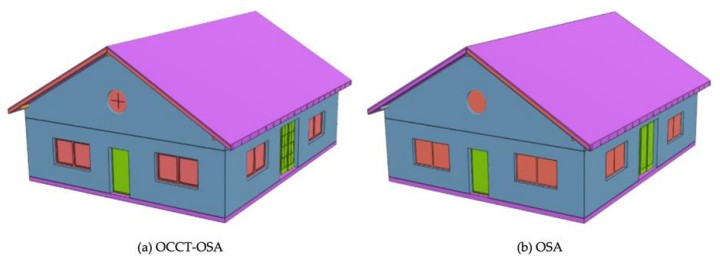

(a) OCCT-OSA          (b) OSA

**Figure 17.** House 1 model generated by (**a**) OCCT-OSA and (**b**) OSA.

(3) The precision problem. Geometric errors are observed in the House 2 model converted by OSA (see Figure 18b). These errors were caused by the precision problem [24], which has been recognized by researchers in the area of computational geometry [74] that can lead geometric algorithms to crash, loop forever, or simply output incorrect results [75]. This problem is rooted in the fact that, in computer systems, infinite real numbers have to be squeezed (or rounded) into the finite floating-point numbers. In the case of IFC, when the modeling precision of BIM models does not meet the requirement, which is the case of House 2, the generated shapefile models are prone to geometric errors, which impairs the quality of the converted BIM models. In this study, OCCT-OSA can effectively handle this problem. In the case of House 2, OCCT-OSA generated the model without geometric errors (see Figure 18a).

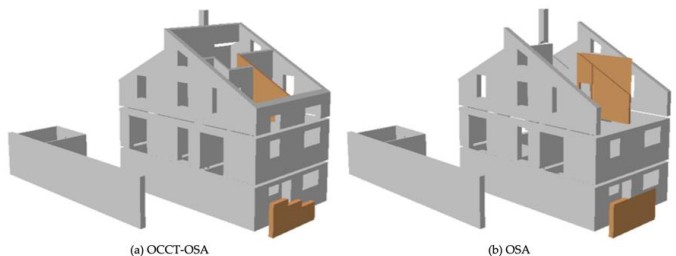

**Figure 18.** Walls generated by (**a**) OCCT-OSA and (**b**) OSA.

### 5.1.2. Comparison between OCCT-OSA and DIA/FME

The latest version of DIA/FME was used for the comparison. Table 10 shows the processing time of OCCT-OSA and DIA/FME, which indicates that OCCT-OSA is more efficient than DIA/FME in processing those models. Even though the conversion can be completed by DIA/FME, the generated models, unfortunately, contain geometric errors, as shown in Figure 19, which indicates that DIA/FME is still not sufficiently reliable.

**Table 10.** Processing time between OCCT-OSA and DIA/FME.

|  | **Processing Time (Seconds)** | | | |
|---|---|---|---|---|
|  | **House 1** | **House 2** | **Smiley** | **Institute** |
| DIA/FME (a) | 25.4 | 25.9 | 118.1 | 136.7 |
| OCCT-OSA (b) | 8.5 | 4.6 | 55.2 | 17.9 |
| Improvement (a-b)/b | 199% | 463% | 114% | 664% |

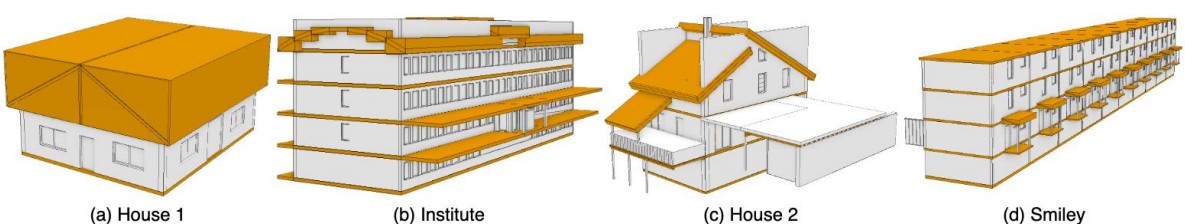

**Figure 19.** Models generated by DIA/FME.

### 5.1.3. Comparative Summary for Methods in IFC-to-Shapefile Conversion

Table 11 presents a comparative summary for methods used in IFC-to-shapefile conversion, including OCCT-OSA, OSA, the method proposed by Isikdag [51], and commercial tools, in terms of geometry conversion, semantic information transfer, the ability to handle precision problem, and if the method is automatic. Compared with previous studies, the proposed OCCT-OSA can automatically convert all types of representations, including B-Rep, swept solid, and CSG/Clipping, and the precision problem can be efficiently handled to guarantee the quality of the converted models.

**Table 11.** Comparative summary for methods in IFC-to-shapefile conversion.

| | Geometry Conversion | | | Semantic Information Transfer | Handling Precision Problem | Automatic |
|---|---|---|---|---|---|---|
| | **B-Rep** | **Swept Solid** | **CSG/Clipping** | | | |
| FME [76], DIA | √ | √ | √* | √* | NA | √ |
| Isikdag [51] | NA | √ | √* | √* | NA | √* |
| OSA [24,28,29] | √ | √ | √ | √* | X | √* |
| OCCT-OSA | √ | √ | √ | √* | √ | √ |

√: solved, √*: partly solved, X: not solved, NA: not assessed.

### 5.2. Validation Using Other Models

Additional BIM models were used to further test OCCT-OSA, including three models from ongoing real projects and six models from the OIMR. Information about these models and data conversion are presented in Table 12. The file size of these models ranges from 70KB to 349.1MB, and the number of building components ranges from 61 to 1906. It can also be noticed that many models do not meet the modeling precision requirement of IFC, but OCCT-OSA can handle all these models with a processing time ranging from 0.55 s to 967.56 s, depending on the size and complexity of these models. Figure 20 presents the converted shapefile models visualized using random colors in ArcScene.

These models were not used in the comparison between the proposed method and OSA, because (a) these models were mainly intended to further assess the performance of the proposed method, and (b) OSA alone cannot process some of these models, as OSA was developed using well-built BIM models (with well-defined modeling precision), such as those from KIT, and may have problems in handling models with geometric errors that are common among models used by real projects [16,25]. This additional validation further justifies the reliability of the proposed method.

### 5.3. Contributions, Implications, Limitations, and Future Work

This study managed to integrate computer graphics techniques, or OCCT, into the conversion of IFC into shapefile; the main contributions of this study are as follows. (1) Exploring and developing an innovative way of using computer graphics techniques at a low level in IFC-to-shapefile conversion. This study creatively used the temporary shape objects (buffers) generated by OCCT for the IFC-to-shapefile conversion, where these buffers were retrieved, reorganized, and preserved in shapefile for use in other applications, despite their original purpose being for visualization. The developed method can also avoid problems in previous studies that are introduced by their inability to efficiently manipulate IFC geometry. These studies used existing tools at process level to build long and sometime complex pipelines for data conversion, which made the conversion process inefficient and prone to errors, such as in studies by Arroyo Ohori et al. [16] and Zhao et al. [59]. (2) Providing a reliable and efficient approach for IFC-to-shapefile conversion. OCCT has been used by many CAD or BIM programs as a 3D modeling kernel, such as the FreeCAD and BIMserver, which makes the proposed method as reliable as these programs. As long as BIM models can be processed and visualized using these programs, they can be converted and used in GIS.

**Table 12.** Additional BIM models used in validation.

| | Model | File Size | Number of Components | Modeling Precision | Time (Seconds) | Data Source |
|---|---|---|---|---|---|---|
| 1 | bridge1.ifc | 70.0 KB | 61 | $1 \times 10^{-4}$ | 0.6 | Project |
| 2 | bridge2.ifc | 845.0 KB | 609 | $1 \times 10^{-5}$ | 8.7 | Project |
| 3 | T18001_Zonghelou.ifc | 2.6 MB | 253 | $1 \times 10^{-2}$ | 7.3 | Project |
| 4 | 20160125OTC-Conference-Center.ifc | 226.6 MB | 1728 | $1 \times 10^{-4}$ | 525.8 | OIMR |
| 5 | 20200109rac_advanced_sample_project.ifc | 103.0 MB | 925 | $1 \times 10^{-2}$ | 244.5 | OIMR |
| 6 | 20191126AZUMA9.ifc | 20.3 MB | 96 | $1 \times 10^{-5}$ | 32.0 | OIMR |
| 7 | 20190228201620_Svaleveien_8_Hus_A.ifc | 17.3 MB | 120 | $1 \times 10^{-5}$ | 38.6 | OIMR |
| 8 | 20160125Autodesk_Hospital_Parking.ifc | 14.3 MB | 1085 | $1 \times 10^{-4}$ | 37.4 | OIMR |
| 9 | 20180731Dubal-Herrera-limpio.ifc | 349.1 MB | 1906 | $1 \times 10^{-5}$ | 967.6 | OIMR |

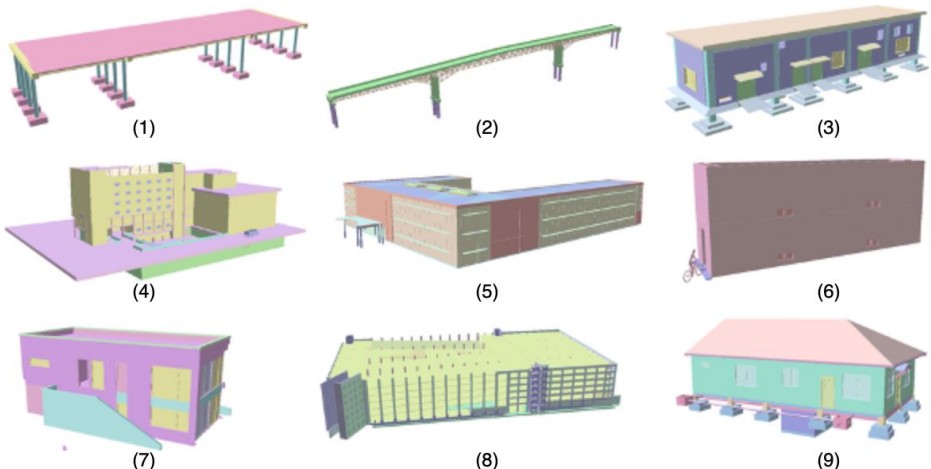

**Figure 20.** Additional models converted by OCCT-OSA.

The implication of this study includes the following aspects. (1) For the area of BIM/GIS integration, BIM models can now be integrated into a GIS environment in an easier way. While the IFC-to-CityGML conversion is still problematic in both geometry conversion and semantics transfer [7,42], this study proposes to use the IFC-to-shapefile conversion path, which makes the BIM-to-GIS data conversion easier and more flexible to realize. In addition, the technique developed in this study for the in-depth use of OCCT can also be used by studies working on IFC-to-CityGML conversion, as these two conversion paths share a common first step, i.e., interpreting IFC geometry. Moreover, the method developed in this study can be the foundation of system-level BIM/GIS integration, where information systems are developed by incorporating both GIS and BIM features. The method has the potential to enable a desktop-based GIS system to directly read and interpret IFC data. (2) For the emerging concept of the smart city and digital twin, where a large number of 3D building models from BIM are required to be processed and used in GIS and the conversion of such a volume of models can consume an enormous amount of time, this fully automatic, reliable, and efficient conversion approach can improve the overall efficiency of projects on these topics. (3) This study can complement other 3D model acquisition techniques for the smart city and digital twin, such as photogrammetry and laser scanning [77]. While existing acquisition techniques capture the surface information of buildings and can be time-consuming and error-prone when reconstructing surface-based building models [78], this study can provide highly detailed solid building models. (4) In a broad sense, the method and new knowledge created in this study for manipulating IFC geometry can be reused by any studies where IFC models are involved.

This study currently mainly focuses on geometry and only preserves the essential semantic information. Even though additional semantic information can be extracted

and linked afterwards with the geometric model in the form of attribute tables, some types of semantic information, such as the topology information of building components, cannot be well maintained. Moreover, the lack of a semantic data schema of shapefile makes the standardization operation more complex. A better solution for semantics transfer is thus needed. Methods using ontology techniques, such as a graph database, should be investigated, as studies have preliminarily demonstrated that the ontology technique is promising in lossless semantic information transfer. Another concern of the proposed method is that it only produces as-is BIM models without the ability to manipulate models, e.g., for the purpose of model simplification, which is important for multi-scale representation in GIS for the sake of reducing storage space and saving rendering power [79]. Lastly, shapefile has been successfully used in this study as a repository for solid building models, but due to its intrinsic limitations on file size and attribute database, as well as its vulnerable data structure, such a format alone may not be suitable for large projects, and a better way of managing such data should be investigated— for example, by using geodatabases. These concerns are the main limitations of this study, which should be addressed in future work.

## 6. Conclusions

This study primarily aims to facilitate the use of BIM models in GIS and focuses on developing a reliable and efficient approach for addressing the fundamental BIM-to-GIS data conversion problem. To achieve this, first, the commonly used IFC-to-CityGML path and the IFC-to-shapefile path were compared in terms of the number and difficulty of data conversion tasks. Second, a more efficient and reliable approach was developed for the IFC-to-shapefile path by integrating a computer graphics technique, i.e., OCCT, in the conversion process. In order to achieve this, the format of the temporary shape object generated by OCCT was investigated and an algorithm was developed to convert OCCT B-Rep to shapefile B-Rep. The proposed method is referred to as OCCT-OSA. Four building models have been used to develop and validate the proposed approach. Nine additional models from real projects or online sources have been used to further test the reliability of the proposed method.

The main findings of this study are as follows. (1) The IFC-to-shapefile conversion is easier and more flexible to realize than the IFC-to-CityGML conversion, in terms of the number and difficulty of involved conversion tasks. (2) Computer graphics techniques can be integrated into IFC-to-shapefile conversion at a low level to improve the efficiency and reliability of BIM-to-GIS data conversion. The developed OCCT-OSA is more efficient than previous methods, and the generated building models contain more geometric details. (3) OCCT-OSA can transform all types of representations, including B-Rep, swept solid, and CSG/Clipping, in a fully automatic manner and effectively handle the precision problem. It can be used to process those not-well-built models that are common in real projects.

The contribution of this study includes the following aspects. (1) A reliable data conversion method for BIM/GIS integration is provided. When the IFC-to-CityGML path for data conversion is still problematic in both geometry conversion and semantics transfer due to mismatches in semantics and modeling method, this study provides an automatic, efficient, and effective data conversion path to ensure that BIM data can effectively flow into GIS. (2) An investigation into OCCT was conducted in depth, and a method for the in-depth use of OCCT in IFC geometry manipulation has been developed, which can also benefit studies on IFC-to-CityGML conversion or any other studies where IFC models are involved, given that OCCT is widely involved in open BIM and CAD.

The proposed method is fundamental, which facilitates the use of BIM models in GIS and supports studies on the smart city and digital twin in a broad sense. The main limitation of this study is that the proposed method currently only generates as-is models without the ability to edit/modify model geometry, while model simplification is important for GIS. In addition, the semantics transfer problem should be better addressed in the future.

**Author Contributions:** Conceptualization, J.Z. and P.W.; methodology, J.Z.; software, J.Z.; validation, J.Z.; writing—original draft preparation, J.Z.; writing—review and editing, J.Z. and P.W.; funding acquisition, P.W. All authors have read and agreed to the published version of the manuscript.

**Funding:** This research was funded by the Australian Research Council, grant number DP180104026.

**Data Availability Statement:** Data available in a publicly accessible repository that does not issue DOIs. Publicly available datasets were analyzed in this study. This data can be found here: https://www.ifcwiki.org/index.php?title=IFC_Wiki (accessed on 11 May 2021), and http://openifcmodel.cs.auckland.ac.nz (accessed on 11 May 2021).

**Acknowledgments:** The authors would like to thank the anonymous reviewers for their comments and suggestions that helped to improve the comprehensiveness and clarity of our paper.

**Conflicts of Interest:** The authors declare no conflict of interest.

## Appendix A  Python Codes for Retrieving Essential Attributes

```
def findSpatialStrcture(buildingElement):
    #determine the type of buildingElement
    if buildingElement.is_a() == 'IfcSpace':
        firstStructure = buildingElement.Decomposes[0].RelatingObject
    elif buildingElement.is_a() == 'IfcSite':
        firstStructure = buildingElement
    else:
        firstStructure = buildingElement.ContainedInStructure[0].RelatingStructure
    if firstStructure.is_a() == 'IfcBuildingStorey':
        storeyEle = findStoreyLevel(firstStructure)
        storeyName = firstStructure.Name
        bldName = firstStructure.Decomposes[0]. RelatingObject.Name.encode('ascii','ignore')
        siteName = firstStructure.Decomposes[0]. RelatingObject.Decomposes[0].RelatingObject.Name
    elif firstStructure.is_a() == 'IfcBuilding':
        storeyEle = None
        storeyName = None
        bldName = firstStructure.Name
        siteName = firstStructure.Decomposes[0].RelatingObject.Name
    elif firstStructure.is_a() == 'IfcSite':
        storeyEle = None
        storeyName = None
        bldName = None
        siteName = firstStructure.Name
    result = [storeyEle, storeyName, bldName, siteName]
    return result
```

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
