# Peer review of "Towards Effective BIM/GIS Data Integration for Smart City by Integrating Computer Graphics Technique"

_remotesensing, doi:10.3390/rs13101889_

Round 1
Reviewer 1 Report
This paper is already in good condition. I only have a few minor recommendations
It is a pity that the paper focuses almost exclusively on ESRI products when it comes to further processing of the data. Is it possible to add information about other tools suitable for handling 3D city models based on shapefiles? Are there any open source tools available that can process the resulting data from your workflow?
In the discussion you compare your methods with others in terms of processing time and file size. What about data integrity and model details?
Reviewer 2 Report
The submitted paper presents a methodology for BIM/GIS integration by using computer graphics techniques. The integration between BIM and GIS is achieved by implementing an IFC-to-shapefile conversion that can be used for integration of BIM and GIS data for smart city and digital twin applications. The presented methodology is tested on a couple of dataset and compared with commercial tools for data conversion.
The topic of the paper is of major importance and several approaches are developed to facilitate BIM and GIS integration. Authors are paying proper credits to related works and the literature review is covering the majority of relevant papers. The paper is clearly structured: the aim is clear since the beginning, the methodology is described in a dedicated paragraph and eventually tested on some reference dataset.
Even if the paper is showing a good level work, I still have some doubts:
- The choice of addressing the topic of BIM/GIS integration by using a IFC-to-shapefile conversion poses some issues. The shapefile makes the conversion easier than a CityGML based approach. However, a standardization issue rises in terms of shapefile structure and attribute definition. The semantic data schema of CityGML helps in standardization while the lack of semantic data schema of shapefile makes the standardization operation more complex. Authors should discuss about this aspect more in depth.
- Concerning semantic information transferring: either a unique shapefile for the entire building is generated or different shapefiles for different building elements (e.g. IfcBeam, IfcColumn) are used? I guess the first one but it is not clear from the text
- The comparison between the proposed method and other conversion tools is carried out mainly in terms of processing time and file size. Discussion on conversion issues for specific object classes is kept only from a visual point of view. I suggest authors to provide a metric to define for each dataset the percentage of building elements correctly generated.
Reviewer 3 Report
I must say that I am very impressed by the work done by the authors. As many other have pointed out, CityGML is a problematic format for many practical reasons, so an alternative BIM->GIS conversion path is very welcome. Even if the Shapefile format is closed, we should recognise that it effectively functions as a de facto standard, with wide software support and many libraries to read/write it. In this sense, it can be argued to be more open than CityGML.
As far as I can tell, the conversion done by the authors works very well, and I am impressed by the high quality figures provided. That being said, fully replicating the authors' work would require access to the authors' source code. In the same vein, I would really encourage the authors to open their code, which would be in the spirit of OCCT and would follow many of the other conversion techniques that the authors have cited.
Also, I would encourage the authors to attempt their conversion with more complex IFC models. The KIT models are very simple, made by researchers and are universally considered as easy to convert. I would really like to see how this methodology works with BIM models from practice, and I think it would be very insightful for the authors as well to be able to see the limits in the current conversion methodology, as well as potential future steps.
Round 2
Reviewer 2 Report
Authors addressed all my previous comments
Author Response
Thank you for all your comments and suggestions.